pattern recognition/bioinformatics

maximal information coefficient, $\chi^2$-test, statistical power, equitability, K-means clustering

**Authors for correspondence:**
Yuan Chen
e-mail: chenyuan0510@126.com
Zheming Yuan
e-mail: zhmyuan@sina.com

# An improved algorithm for the maximal information coefficient and its application

Dan Cao[1,2], Yuan Chen[1], Jin Chen[1], Hongyan Zhang[1] and Zheming Yuan[1]

[1]Hunan Engineering and Technology Research Centre for Agricultural Big Data Analysis and Decision-making, Hunan Agricultural University, Changsha 410000, People's Republic of China
[2]Orient Science and Technology College of Hunan Agricultural University, Changsha 410000, Hunan, People's Republic of China

ZY, 0000-0002-7444-8743

The maximal information coefficient (MIC) captures both linear and nonlinear correlations between variable pairs. In this paper, we proposed the BackMIC algorithm for MIC estimation. The BackMIC algorithm adds a searching back process on the equipartitioned axis to obtain a better grid partition than the original implementation algorithm ApproxMaxMI. And similar to the ChiMIC algorithm, it terminates the grid search process by the $\chi^2$-test instead of the maximum number of bins $B(n, \alpha)$. Results on simulated data show that the BackMIC algorithm maintains the generality of MIC, and gives more reasonable grid partition and MIC values for independent and dependent variable pairs under comparable running times. Moreover, it is robust under different $\alpha$ in $B(n, \alpha)$. MIC calculated by the BackMIC algorithm reveals an improvement in statistical power and equitability. We applied (1-MIC) as the distance measurement in the K-means algorithm to perform a clustering of the cancer/normal samples. The results on four cancer datasets demonstrated that the MIC values calculated by the BackMIC algorithm can obtain better clustering results, indicating the correlations between samples measured by the BackMIC algorithm were more credible than those measured by other algorithms.

## 1. Introduction

Correlation analysis has important applications in data mining, such as disease diagnosis [1,2], public management [3,4] and financial market analysis [5,6]. Since the maximal information coefficient (MIC) [7] was proposed as a type of nonlinear correlation measurement, it has been widely studied and applied because of its generality and approximate equitability [8–11]. In MIC, if there is a

**Figure 1.** Grid frequency distribution of the three algorithms for independent variable pairs. *y*-axis and *x*-axis are the number of bins. Data size $n = 100$, 1000 replicates.

correlation between a paired variable, a grid can be drawn on the scatterplot of these two variables to encapsulate this correlation. MIC is not affected by the distribution and correlation type of the variable pairs [7]

Given a paired variable $(X, Y)$, $X \in R^n$, $Y \in R^n$, MIC of $X$ and $Y$ is defined as

$$\text{MIC}(X, Y) = \max_{n_x \times n_y \leq B(n, \alpha)} \left\{ \frac{\max_G (I_G(X, Y))}{\log_2 \min(n_x, n_y)} \right\},$$

where $n_x$ and $n_y$ are the number of bins on the *x*-axis and *y*-axis, respectively. $G$ represents a $n_x \times n_y$ grid on $(X, Y)$, $I_G(X, Y)$ denotes the mutual information under the grid $G$. $B(n, \alpha)$ is a function of data size $n$ and is equal to $n^\alpha$ $(0 < \alpha < 1)$, which limits the maximum number of bins. $\log_2 \min(n_x, n_y)$ is a normalization term to ensure MIC in the range of 0 to 1. MIC converges to 0 as data size $n \to \infty$ when $X$ and $Y$ are statistically independent; the MIC increases as the correlation between $X$ and $Y$ strengthens.

The calculation of MIC is computationally intensive. To estimate MIC, we need to search for the 'optimal grid' on the scatterplot of $X$ and $Y$ to separate the 'blank space' and 'data points' by using the least number of bins possible. With the same number of bins, the grid improves as the MIC increases. Several algorithms can be used for the approximate estimation of MIC. For ApproxMaxMI (AppMIC, downloaded from http://www.exploredata.net/Downloads/MINE-Application), proposed by Reshef *et al.* [7], one axis (the *y*-axis or the *x*-axis) is equipartitioned (an *equipartition* is a partition into either rows or columns such that each row/column contains the same number of data points), while the other axis is partitioned by dynamic programming algorithm under the limitation of $B(n, \alpha)$ ($\alpha$ is often set to 0.55 or 0.6; in this paper, $\alpha$ is 0.6). However, AppMIC algorithm features two problems. First, the equipartition of one axis is neither sufficient nor necessary for MIC estimation [12]. Second, the MIC values are usually obtained when the number of bins $n_x \times n_y$ reaches $B(n, \alpha)$, so the generality of MIC is closely related to $B(n, \alpha)$. If a low $B(n, \alpha)$ is set, then the MIC can only capture simple correlation patterns; by contrast, a high $B(n, \alpha)$ will cause a non-zero score even for independent variables [7]. To solve this problem, Chen *et al.* [13] proposed the ChiMIC algorithm (downloaded from https://github.com/chenyuan0510/Chi-MIC), in which one axis is equipartitioned, and the partition of other axis is terminated by the $\chi^2$-test. To avoid the number of bins on the equipartitioned axis always reaching $B(n, \alpha)/2$, the ChiMIC algorithm replaces $\log_2 \min(n_x, n_y)$ in the MIC definition with $\log_2 n_{\text{equ}}$, where $n_{\text{equ}}$ is the number of bins on the equipartitioned axis and $\log_2 n_{\text{equ}} \geq \log_2 \min(n_x, n_y)$. However, whether the normalization term can be modified remains unclear, and the equipartition restriction remains unresolved in the ChiMIC algorithm.

In this study, we proposed an improved approximation algorithm called BackMIC for MIC estimation. This algorithm adds a searching back process on the equipartitioned axis to remove the restriction of equipartition and control the search process based on the $\chi^2$-test for both the *y*- and *x*-axes. Results on simulated and real data demonstrated that the BackMIC algorithm exhibits better performance in measuring the correlations between independent and dependent variable pairs compared with the AppMIC and ChiMIC algorithms.

## 2. Results and discussion

### 2.1. Comparison of grids and estimated MICs for independent variable pairs

The expected grid for independent variable pairs is $2 \times 2$ [13]. Figure 1 shows the grid frequency distribution obtained by the AppMIC, ChiMIC and BackMIC algorithms for computing the MIC values of independent variable pairs at 1000 repetitions. When data size $n = 100$ and $B(n, \alpha) = 16$,

**Table 1.** MICs estimated by different algorithms for various data sizes. The MIC values are represented as the average value ± the standard deviation of 1000 repetitions.

| data size | AppMIC | ChiMIC | BackMIC |
| --- | --- | --- | --- |
| 250 | 0.1813 ± 0.0234 | 0.0390 ± 0.0284 | 0.0556 ± 0.0180 |
| 500 | 0.1545 ± 0.0152 | 0.0188 ± 0.0165 | 0.0342 ± 0.0104 |
| 1000 | 0.1051 ± 0.0103 | 0.0119 ± 0.0092 | 0.0204 ± 0.0065 |
| 2000 | 0.0730 ± 0.0070 | 0.0066 ± 0.0053 | 0.0119 ± 0.0032 |
| 4000 | 0.0490 ± 0.0040 | 0.0041 ± 0.0035 | 0.0069 ± 0.0018 |
| 10 000 | 0.0383 ± 0.0024 | 0.0020 ± 0.0013 | 0.0032 ± 0.0008 |
| 20 000 | 0.0221 ± 0.0011 | 0.0010 ± 0.0009 | 0.0018 ± 0.0005 |

**Table 2.** Thirteen functional correlations.

| functions | $X$ | $f(X)$ |
| --- | --- | --- |
| line | [0, 1] | $X$ |
| parabolic | [−0.5, 0.5] | $4X^2$ |
| cubic | [−1.3, 1.1] | $4X^3 + X^2 − 4X$ |
| exponential | [0, 10] | $2^X$ |
| non-Fourier freq [low] | [0, 1] | $\cos(4\pi X)$ |
| non-Fourier freq [medium] | [0, 1] | $\cos(8\pi X)$ |
| non-Fourier freq [high] | [0, 1] | $\cos(12\pi X)$ |
| linear + period freq [low] | [0, 1] | $(1/5)\sin(4(2X − 1)) + (11/10)(2X − 1)$ |
| linear + period freq [medium] | [0, 1] | $\sin(10\pi X) + X$ |
| linear + period freq [high] | [0, 1] | $\dfrac{1}{10}\sin(10.6(2X − 1)) + \dfrac{11}{10}(2X − 1)$ |
| varying freq [low] | [0, 1] | $\cos(4\pi X (1 + X))$ |
| varying freq [medium] | [0, 1] | $\cos(8\pi X (1 + X))$ |
| varying freq [high] | [0, 1] | $\cos(12\pi X (1 + X))$ |

almost all the grids of the AppMIC algorithm were concentrated in $2 \times 8$ and $8 \times 2$. Most grids of the ChiMIC algorithm were concentrated in $a \times b$, where $a \leq 5$ and $b \leq 5$. By contrast, almost all the grids of the BackMIC algorithm were concentrated in $2 \times 2$, $2 \times 3$ and $3 \times 2$; the most frequent grid was $2 \times 2$. The grids of the BackMIC algorithm were the closest to the expected one.

The MIC of independent variable pairs should converge to 0 as data size $n \to \infty$ [7]. Table 1 shows the MIC values estimated by the three algorithms (i.e. AppMIC, ChiMIC and BackMIC) for different data sizes. BackMIC was similar to ChiMIC without expanding the normalization term, which was closer to zero than AppMIC under the same data size.

## 2.2. Comparison of grids and estimated MICs for dependent variable pairs

The MIC of variable pairs with noiseless functional correlations should be 1 [7]. We used the BackMIC algorithm to calculate the MIC values of 13 pairs of noiseless functional correlations (table 2). All MIC values were 1, indicating that the BackMIC algorithm maintained the generality of MIC.

For the 'T'-type dataset, the expected grid and MIC were $3 \times 3$ and 0.2561, respectively (figure 2$a$). The grids of the AppMIC, ChiMIC and BackMIC algorithms were $14 \times 3$, $3 \times 3$ and $3 \times 3$, respectively, and the estimated MICs were 0.2457, 0.1808 and 0.2561, respectively (figure 2$b$–$d$). Only the grid and estimated value of the BackMIC algorithm were in line with expectations. For the chequerboard dataset, the expected grid and MIC were $5 \times 5$ and 0.3835, respectively (figure 2$e$). The grids of the AppMIC, ChiMIC and BackMIC algorithms were $9 \times 5$, $5 \times 6$ and $5 \times 6$, respectively, and the estimated

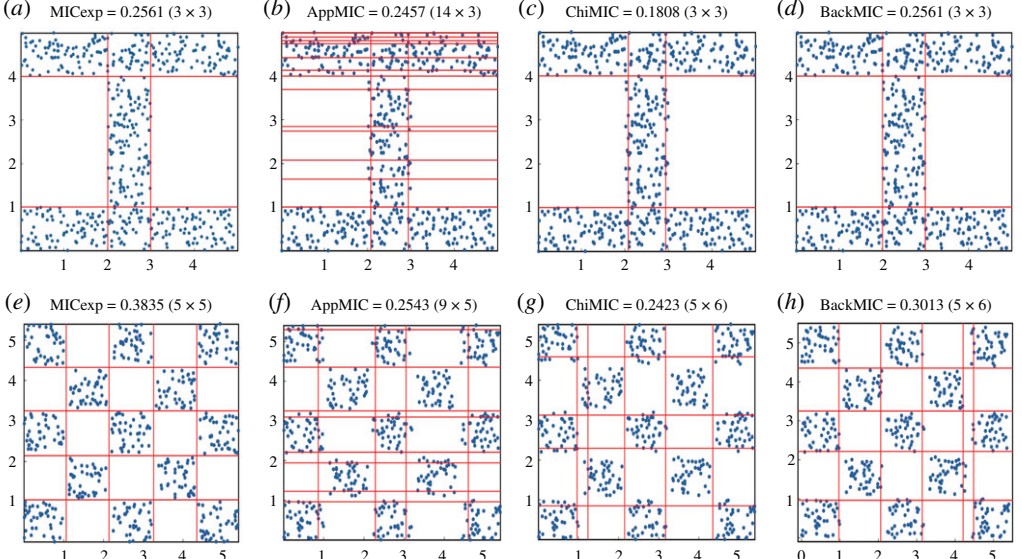

**Figure 2.** Comparison of the grids and estimated MICs for the 'I'-type and chequerboard datasets. Data size $n = 520$; each square contains 40 data points.

MICs were 0.2543, 0.2423 and 0.3013, respectively (figure 2f–h). The grid and estimated MIC of the BackMIC algorithm were the closest to expectations relative to those of the other algorithms.

The actual MIC values of noisy linear, parabolic and sinusoidal correlations were difficult to determine, as shown in figure 3. We compared the three algorithms under the criterion that 'with the same number of bins, the larger the MIC values, the better the grid'. The AppMIC algorithm was excluded from the comparison because of its excessive bins (all reached $2 \times 21$ for data size $n = 500$ and B($n$, $\alpha$) = 42; figure 3a, d and g). A comparison of the MIC values of the three functional correlations obtained by the ChiMIC and BackMIC algorithms revealed that the latter always achieved higher MIC values with the same or fewer number of bins than the former. Thus, the grid and estimated MICs obtained by the BackMIC algorithm were more reasonable than those by the ChiMIC algorithm; no axis was equipartitioned.

## 2.3. Comparison of robustness

The correlation strength of a given variable pair should be certain. However, as shown in figure 4, AppMICs varied with the change of $\alpha$ in B($n$, $\alpha$) for noisy linear, parabolic and sinusoidal correlations, because more bins generally result in larger MIC values. BackMICs remained almost constant because the $\chi^2$-test was used instead of B($n$, $\alpha$) to terminate grid optimization. Therefore, the BackMIC algorithm was more robust in measuring the correlation between variables than the AppMIC algorithm. The variation in ChiMICs was not obvious either; however, ChiMICs were always lower than BackMICs due to the equipartition restriction and harsh normalization term in the ChiMIC algorithm.

## 2.4. Comparison of statistical power

Statistical power refers to the probability of correctly accepting the alternative hypothesis in the hypothesis test [14]. As the statistical power increases, the probability of making type II error decreases [15,16].

For the null hypothesis of statistical independence, for each dataset, statistical power is computed on the dependent variable pairs as well as on independent variable pairs; the statistical power of each statistic is defined as the fraction of dependent variable pairs yielding a statistic value greater than 95% (significance level is 0.05) of the values yielded by the independent variable pairs [13]. The statistical power of AppMIC, ChiMIC and BackMIC for the above five functional correlations (figures 2 and 3) at different noise amplitudes is shown in figure 5. The power of BackMIC was significantly higher than those of AppMIC and ChiMIC.

## 2.5. Comparison of equitability

If a statistic assigns similar scores to equally noisy correlations of different types, then the statistic has the property of equitability [17]. Equitability allows us to specify a threshold correlation strength below

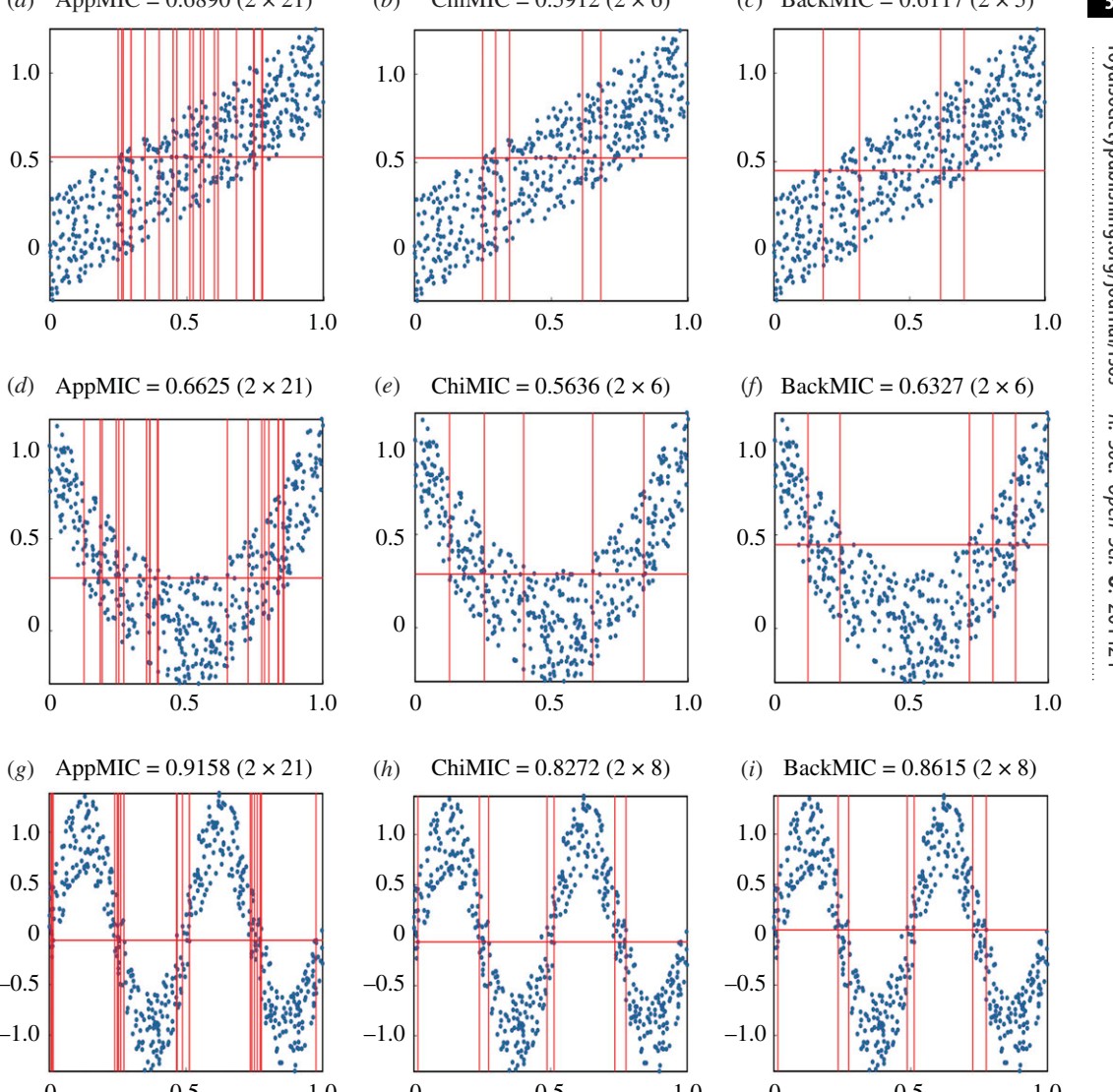

**Figure 3.** Comparison of the grids and estimated MICs for linear (a–c), parabolic (d–f) and sinusoidal (g–i) correlations. The data points are simulated from $Y = f(X) + \zeta$, where $f(X) = X$, $4(X - 0.5)^2$ and $\sin(4\pi X)$, respectively. $X$ and $\zeta$ are random variables drawn uniformly from (0, 1) and (−0.3, 0.3), respectively; data size $n = 500$.

which we are uninterested and search for correlations whose strength is greater than the threshold [18]. Perfect equitability does not exist [19]. We tested the approximate equitability of AppMIC, ChiMIC and BackMIC on the basis of 13 functional correlations listed in table 2. For each correlation, we analyse the equitability by generating a noiseless data sequence with a data size $n = 500$ and 301 data series with noise $\varepsilon$ added to $f(X)$, where $\varepsilon$ is a uniformly distributed random variable from $-b$ to $b$, and $b$ denotes the noise levels selected from [0, 3] with step size of 0.01. The results confirmed that the approximate equitability of BackMIC was better than those of AppMIC and ChiMIC (figure 6).

## 2.6. Comparison of computational cost

We compared the computational time of the AppMIC, ChiMIC and BackMIC algorithms by using different data sizes for two types of variable pairs (the independent variable pairs and the parabolic functional variable pairs with noise level of 0.5). The results in table 3 showed that the ChiMIC algorithm ran faster than the AppMIC algorithm because the former used the $\chi^2$-test to terminate grid optimization earlier than the latter. The running time of the BackMIC algorithm was almost twice as much as that of the ChiMIC algorithm, because the BackMIC algorithm added a searching back process for an optimal partition on the originally equipartitioned axis compared with the ChiMIC algorithm. Compared with the AppMIC algorithm, the BackMIC algorithm was slower when the data size was small. As the data

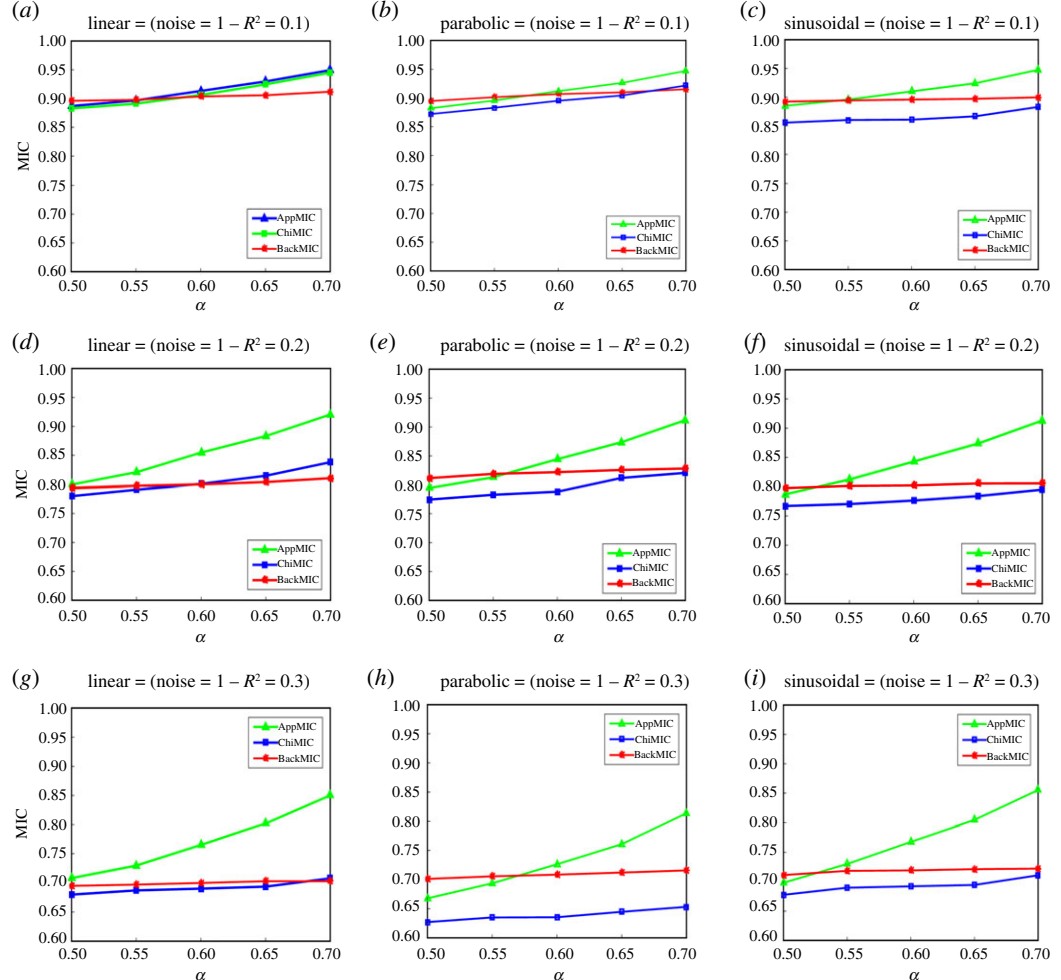

**Figure 4.** Comparison of AppMIC, ChiMIC and BackMIC for linear, parabolic and sinusoidal correlations at different $\alpha$ in B($n$, $\alpha$). $R^2$ in 'noise' is the squared Pearson correlation coefficient of $f(X)$ and $Y$, where $f(X)$ is the same as in figure 3 and $Y = f(X) + \zeta$. For linear and parabolic correlations, $\zeta$ is drawn uniformly at noise levels 0.2, 0.35 and 0.5, respectively; for sinusoidal correlation, $\zeta$ is drawn uniformly at noise levels 0.6, 0.9 and 1.2, respectively. The MIC values were the average of 500 repetitions with data size $n = 500$.

size increased, the BackMIC algorithm was able to catch up with the AppMIC algorithm. In terms of independent variable pairs, the BackMIC algorithm ran faster than the AppMIC algorithm.

## 2.7. Comparison of AppMIC, ChiMIC and BackMIC applied in clustering for cancer classification

Given that cancer samples and normal samples have different gene expression levels [20], clustering algorithms such as K-means locate samples into different clusters on the basis of the similarity (distance) between samples; these clusters can then be used for cancer classification [21,22]. To evaluate the performance of MIC obtained by the AppMIC, ChiMIC and BackMIC algorithms, we replaced the Euclidean distance with (1-MIC) as the distance measurement between two samples in the K-means algorithm. Cancer gene expression datasets GSE37023, GSE29272 and GSE35602 were used in our work. Table 4 shows that the purity and Rand index (RI) of clustering results based on (1-BackMIC) were higher than those based on (1-AppMIC) and (1-ChiMIC), and they were even better than those based on Euclidean distance in the original K-means.

# 3. Datasets

## 3.1. Simulated data

Five dependent correlations used in figures 2 and 5 are defined in table 5. $(X_0, Y_0)$ and $(X_1, Y_1)$ are pairs of random variables drawn uniformly from the solid squares of a 'T'-type pattern and $5 \times 5$ chequerboard

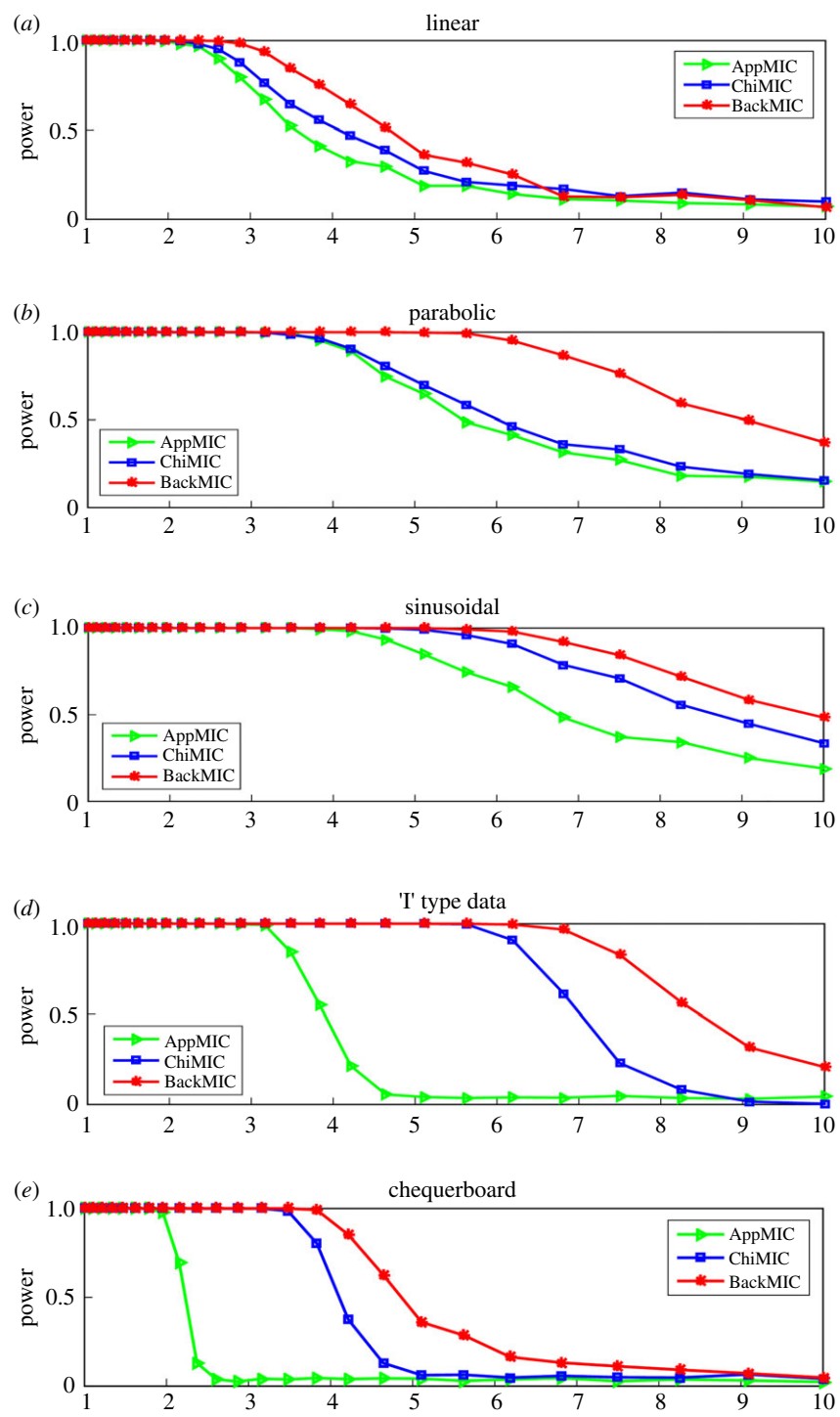

**Figure 5.** Statistical power of AppMIC, ChiMIC and BackMIC at different noise amplitudes. The statistical power was estimated via 500 simulations with data size $n = 500$.

[18], where each square has sides with a length of 1. In figure 2, $a$ and $\eta$ are equal to 0 in 'I'-type pattern and $5 \times 5$ chequerboard. In figure 5, the noise aptitude $a$ for the statistical power calculation is 25 noise amplitudes with logarithmic distribution from 1 to 10; X, $\xi$ and $\eta$ are random variables drawn from the normal distribution $N(0,1)$.

In figure 4, the noise added to linear, parabolic and sinusoidal functional correlations is defined as $Y = f(X) + \text{noise\_level} \times (2\text{rand}(n,1) - 1)$, where rand$(n,1)$ is used to generate $n$ uniformly distributed numbers in [0, 1].

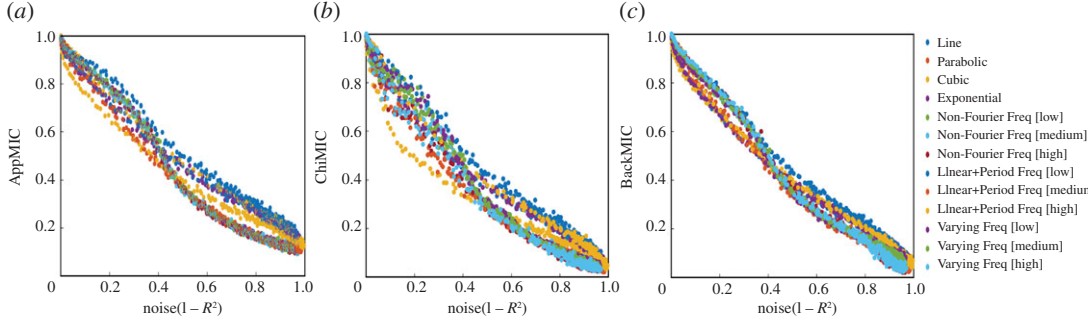

**Figure 6.** Equitability of AppMIC, ChiMIC and BackMIC. These values are the average of 500 repetitions with data size $n = 500$.

**Table 3.** Running time for calculating MICs for different data sizes. The corresponding time was represented as the average value ± standard deviation over 100 repetitions on a Windows 7 64-bit operating system (RAM: 8.00 GB, CPU: 2.60 GHz).

| data size | independent variable pair | | | parabolic function (noise level = 0.5) | | |
|---|---|---|---|---|---|---|
| | AppMIC | ChiMIC | BackMIC | AppMIC | ChiMIC | BackMIC |
| 250 | $0.0184 \pm 3 \times 10^{-3}$ | $0.0130 \pm 1 \times 10^{-3}$ | $0.0240 \pm 3 \times 10^{-3}$ | $0.0226 \pm 3 \times 10^{-3}$ | $0.0147 \pm 3 \times 10^{-3}$ | $0.0299 \pm 4 \times 10^{-3}$ |
| 500 | $0.0401 \pm 0.01$ | $0.0357 \pm 0.01$ | $0.0683 \pm 0.02$ | $0.0605 \pm 0.01$ | $0.0508 \pm 0.01$ | $0.1001 \pm 0.01$ |
| 1000 | $0.3172 \pm 0.04$ | $0.1026 \pm 0.01$ | $0.1774 \pm 0.04$ | $0.2225 \pm 0.03$ | $0.2114 \pm 0.03$ | $0.3856 \pm 0.03$ |
| 2000 | $1.4335 \pm 0.10$ | $0.6163 \pm 0.13$ | $1.0452 \pm 0.20$ | $0.9633 \pm 0.10$ | $0.8694 \pm 0.07$ | $1.5492 \pm 0.16$ |
| 4000 | $6.7017 \pm 0.41$ | $2.4343 \pm 0.38$ | $4.2755 \pm 0.49$ | $3.7833 \pm 0.35$ | $3.0101 \pm 0.15$ | $4.7625 \pm 0.48$ |

**Table 4.** Performance of different distance measurements for K-means clustering. Bold items highlight the highest prediction accuracy.

| datasets | methods | K-means (Euclidean) | K-means (1-AppMIC) | K-means (1-ChiMIC) | K-means (1-BackMIC) |
|---|---|---|---|---|---|
| GSE37023[1] | purity | 0.7815 | 0.8940 | 0.9007 | **0.9338** |
| | RI | 0.6539 | 0.8061 | 0.8167 | **0.8780** |
| GSE37023[2] | purity | 0.7385 | 0.8769 | 0.8000 | **0.9077** |
| | RI | 0.6077 | 0.8048 | 0.6750 | **0.8298** |
| GSE29272 | purity | 0.5075 | 0.9366 | 0.9478 | **0.9627** |
| | RI | 0.4982 | 0.8807 | 0.9073 | **0.9141** |
| GSE35602 | purity | **1.0000** | 0.9198 | 0.9412 | **1.0000** |
| | RI | **1.0000** | 0.8342 | 0.8859 | **1.0000** |

## 3.2. Real datasets

We used GSE37023 [23], GSE29272 [24] and GSE35602 [25] to verify the reliability of BackMIC, as described in table 6. In GSE37023, two datasets from platforms GPL96 and GPL97 were used respectively, denoted as GSE37023[1] and GSE37023[2]. In GSE35602, only the dataset from the platform GPL6480 was used. All the datasets were obtained from the Gene Expression Omnibus database (https://www.ncbi.nlm.nih.gov/geo/). Probe IDs were converted to gene symbols according to GEO platform (GPL). If several probes were mapped to the same gene symbol, their average value was taken as the expression value of this gene. An implementation of the BackMIC algorithm can be downloaded at https://github.com/Caodan82/BackMIC.

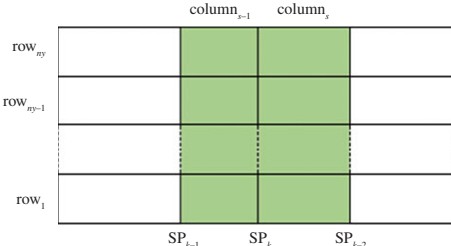

**Figure 7.** Instruction for the calculation of the $\chi^2$ statistic [13].

**Table 5.** Five dependent datasets simulated for statistical power estimation.

| dependent correlations | $X$ | $Y$ |
|---|---|---|
| 'I'-type data | $X_0$ | $Y_0 + a\eta$ |
| chequerboard | $X_1$ | $Y_1 + a\eta$ |
| linear | $\xi$ | $X + a\eta$ |
| parabolic | $\xi$ | $4(X - 0.5)^2 + a\eta$ |
| sinusoidal | $\xi$ | $\sin(4\pi X) + a\eta$ |

**Table 6.** Four cancer gene expression datasets.

| GEO accession | no. genes | no. classes | description |
|---|---|---|---|
| GSE37023[1] | 13 515 | 2 | tumours (112) and non-malignant (39) |
| GSE37023[2] | 11 349 | 2 | tumours (29) and non-malignant (36) |
| GSE29272 | 13 515 | 2 | cancer (134) and normal (134) |
| GSE35602 | 19 595 | 2 | cancer (26) and normal (8) |

# 4. Methods

## 4.1. $\chi^2$-test to terminate grid optimization

The BackMIC algorithm uses the $\chi^2$-test to terminate grid optimization, which is similar to the ChiMIC algorithm. Given an optimal segment point, if the $p$-value of the $\chi^2$-test for the optimal segment point is lower than a given threshold (here, threshold = 0.01), the segment point is valid and the BackMIC algorithm continues to search for the next optimal segment point. Otherwise, the BackMIC algorithm stops its searching process.

We take the $k$th optimal segment point $\text{SP}_k$ on the $x$-axis as an example. Suppose that the $y$-axis is partitioned into $n_y$ bins, and $\text{SP}_k$ is between $\text{SP}_{k-2}$ and $\text{SP}_{k-1}$. The $x$-axis is partitioned into $(s-1)$th and $s$th columns (figure 7). The coloured $n_y \times 2$ contingency table in figure 7 is called the detection area of $\text{SP}_k$ for the $\chi^2$-test. The $\chi^2$ statistic is defined as follows [13]:

$$\chi^2_{\text{SP}_k} = \sum_{j=1}^{n_y} \sum_{i=s-1}^{s} \frac{(n_{j,i} - n_{*,i} n_{j,*}/N_d)^2}{n_{*,i} n_{j,*}/N_d},$$

where $n_{j,i}$ is the number of data points in the bin of row$_j$ and column$_i$, $n_{*,i}$ is the number of data points in the bins of column$_i$, $n_{j,*}$ is the number of data points in the bins of row$_j$ of the detection area, and $N_d$ is the total number of data points in the bins of the detection area. If $n_y = 2$, the $\chi^2$ statistic needs to be corrected according to the following formula [26]:

$$\chi^2_{\text{SP}_k} = \sum_{j=1}^{n_y} \sum_{i=s-1}^{s} \frac{(|n_{j,i} - n_{*,i} n_{j,*}/N_d| - 0.5)^2}{n_{*,i} n_{j,*}/N_d}.$$

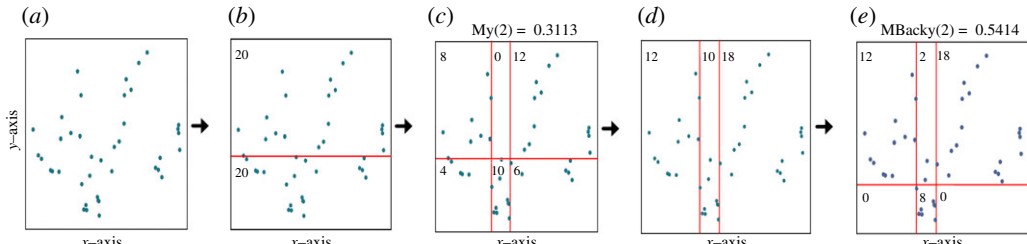

**Figure 8.** Simulation process of the BackMIC algorithm. (*a*) The scatterplot of a paired variable with data size $n = 40$; (*b*) equipartition the *y*-axis into 2 bins; (*c*) search for the optimal partition on the *x*-axis to obtain the largest normalized mutual information $M_y(2) = 0.3113$; (*d*) fix the partition of the *x*-axis obtained in (*c*); (*e*) search back for the optimal partition on the *y*-axis to obtain the largest normalized mutual information $MBack_y(2) = 0.5414$.

## 4.2. BackMIC algorithm

The BackMIC algorithm involves two phases to obtain the optimal partition on the scatterplot of a variable pair (see algorithm 1). First, based on an equipartition of $n_y$ bins on one axis (we take the *y*-axis as example), the BackMIC algorithm locates an optimal partition of the *x*-axis through the dynamic programming algorithm to achieve the largest normalized mutual information under the restriction of the $\chi^2$-test, which is similar to the ChiMIC algorithm [13]. Unlike the AppMIC and ChiMIC algorithms, the BackMIC algorithm fixes the partition of the *x*-axis obtained in the previous step and searches back for the optimal partition of the *y*-axis instead of equipartitioning. Therefore, the BackMIC algorithm controls the bins of both the *y*- and *x*-axes by the $\chi^2$-test and obtains unequipartitioning for both the *y*- and *x*-axes. For $n_y = 2$, the simulation process of the BackMIC algorithm is shown in figure 8. From the results we find that, compared with equipartitioning of the *y*-axis, when the *y*-axis is unequipartitioned, larger normalized mutual information can be obtained (0.5414 versus 0.3113).

**Algorithm 1 BackMIC algorithm**

| | |
|---|---|
| **Input: $(X, Y) = \{(x_1, y_1), (x_2, y_2), …, (x_n, y_n)\}$ is a pair of ordered variables, B=$n^{0.6}$, threshold=0.01** | |
| 1 | for $n_y$ =2 to ceil(B/2) |
| 2 | EquipartitionOf$y$ into $n_y$ rows; PartionOf$x$ ←$\varphi$; BackPartionOf$y$←$\varphi$ |
| 3 | for $n_x$ =2 to $n$ |
| 4 | search for SegPointOf$x_{nx}$ maximizing $I_G(X, Y)/\log_2 \min(n_x, n_y)$; |
| 5 | Obtain $\chi^2_{SegPointOfxnx}$ and $p$-value$_{SegPointOfxnx}$; |
| 6 | if $p$-value$_{SegPointOfxnx}$ > threshold **then** exit for **end if**; |
| 7 | PartionOf$x$ ← PartionOf$x$ ∪{ SegPointOf$x_{nx}$ }; |
| 8 | **end for** |
| 9 | $M_y(n_y)$←$I_G(X, Y)/\log_2 \min($length(PartionOf$x$)+1, $n_y)$; |
| 10 | for $s$ =2 to $n$ |
| 11 | Fix PartionOf$x$, search for SegPointOf$y_s$ maximizing $I_G (X, Y)/\log_2 \min (s,$length(PartionOf$x$)+1); |
| 12 | obtain $\chi^2_{SegPointOfys}$ and $p$-value$_{SegPointOfys}$; |
| 13 | if $p$-value$_{SegPointOfys}$ > threshold **then** exit for **end if**; |
| 14 | BackPartionOf$y$ ← BackPartionOf$y$ ∪ { SegPointOf$y_s$ }; |
| 15 | **end for** |
| 16 | $MBack_y(n_y)$←$I_G(X,Y)/ \log_2 \min ($length(BackPartionOf$y$)+1, length(PartionOf$x$)+1); |
| 17 | **end for** |
| 18 | Switch the axes, repeat the above steps, and obtain $M_x$ and $MBack_x$. |
| **Output** | **MIC = max $\{M_x, MBack_x, M_y, MBack_y\}$** |

## 4.3. K-means clustering algorithm

Suppose there are $M$ samples with $K$ classes in a dataset, $C = \{c_1, c_2, …, c_K\}$ is the sample set of real classifications, and $c_i$ is the sample set of the $i$th class (given that four gene expression datasets are of binary class, $K$ is 2 here). The K-means clustering algorithm is performed using Matlab scripts downloaded at http://people.revoledu.com/kardi/tutorial/kMean/download.htm [27], which proceeds by randomly selecting $K$ initial clustering centres and then assigns each sample to the nearest clustering centre [28]. For each dataset, we used the top 1000 genes with the largest variance to calculate the distance between samples. Suppose that the clustering results of K-means is $\Omega = \{\omega_1, \omega_2, … ,\omega_K\}$, where $\omega_i$ is set of the $i$th cluster.

Two commonly used evaluation criteria of clustering algorithms, namely, purity and RI, were used in this paper. Purity is the proportion of correctly clustered samples in total samples, and it can be calculated by [29]

$$\text{Purity} = \frac{1}{M} \sum_i \max_j |c_i \cap \omega_j|.$$

RI refers to the proportion of concordant sample pairs in the total number of sample pairs [30]. A is the number of sample pairs placed in the same group in both C and Ω, and B is the number of sample pairs placed in different groups in both C and Ω. RI is defined as follows [31]:

$$\text{RI} = \frac{A + B}{M(M - 1)}.$$

## 5. Conclusion

In this paper, we introduced the BackMIC algorithm for better MIC estimation. The BackMIC algorithm added a searching back process to obtain an optimal partition for the equipartitioned axis, making it more likely to obtain the true MIC value. Meanwhile, the BackMIC algorithm used the $\chi^2$-test to ensure that each introduced optimal segment point can significantly increase the MIC value. This effectively avoided unreasonable grid refinement and made MIC value independent of B($n, \alpha$) which improved the robustness of MIC.

The results on simulation data showed that, compared with the AppMIC and ChiMIC algorithms, the BackMIC algorithm can effectively reduce the MIC value of independent variable pairs without expanding the normalization term in MIC definition; if there is a functional correlation between variable pairs, the MIC calculated by the BackMIC algorithm is equal to 1, maintaining the generality of MIC; if there is noisy correlation between variable pairs, the BackMIC algorithm usually obtained larger MIC values with less bins; moreover, the statistical power and equitability of MIC calculated by the BackMIC algorithm are better. When applying the (1-MIC) value as the distance measurement between cancer samples and normal samples in K-means algorithm, experiments on four cancer datasets showed that the MIC values calculated by the BackMIC algorithm can obtain better clustering results. All evidence verified that the BackMIC algorithm improves MIC estimation.

Data accessibility. GSE37023 is at https://www.ncbi.nlm.nih.gov/geo/query/acc.cgi?acc=GSE37023. GSE29272 is at https://www.ncbi.nlm.nih.gov/geo/query/acc.cgi?acc=GSE29272. GSE35602 is at https://www.ncbi.nlm.nih.gov/geo/query/acc.cgi?acc=GSE35602. An implementation of the BackMIC algorithm is stored in GitHub: https://github.com/Caodan82/BackMIC, and has been archived within the Zenodo repository: https://zenodo.org/record/4280173#.X7X3z_nDpw8 (https://doi.org/10.5281/zenodo.4280173).
Authors' contributions. D.C., Y.C. and Z.Y. contributed to the design of algorithm, D.C. carried out the analysis and drafted the manuscript, J.C. and H.Z. helped draft the manuscript, all authors reviewed and gave final approval for submission.
Competing interests. We have no competing interests.
Funding. This research was supported by the National Natural Science Foundation of China (grant no. 61701177), the Scientific Research Foundation of Education Office of Hunan Province, China (grant nos. 17A096 and 18A105).

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
