## [Peer Review File · Royal Society Open Science]

Review History

RSOS-201424.R0 (Original submission)

Review form: Reviewer 1

Is the manuscript scientifically sound in its present form?

Yes

Are the interpretations and conclusions justified by the results?

Yes

Is the language acceptable?

Yes

Do you have any ethical concerns with this paper?

No

Have you any concerns about statistical analyses in this paper?

No

Recommendation?

Accept with minor revision (please list in comments)

Comments to the Author(s)

The author developed BackMIC algorithm for MIC estimation. It removed the equipartition restriction in grid partition, and terminated the grid search process by the chi-square test. The results showed that BackMIC algorithm gives more accurate MIC values, and improves the robustness, statistical power and equitability of MIC. The application of BackMIC to samples clustering in cancer classification is also valuable.

There are still some problems to be noticed:

1. The BackMIC algorithm removes the equipartition restriction of one axis. What is the axis?
2. the calculation of MIC needs more detailed explanation.
3. At present, many researches are focused on very large data sets, but when comparing the time complexity, the maximum data size is 4000, and the advantages are not obvious. It is recommended to compare on larger data sets.
4. There are a few errors. 1) Where is Eq. (2) in line 32? 2) Figure 2h is a mistake, the author mentioned in line 56 that the grid of BackMIC algorithms for the checkboard data is 5×6 , however, the grid in Figure 2h is 5×5 .
5. Pay attention to English grammar and sentence structure to ensure that the paper can be better understanding.

Review form: Reviewer 2 (Janne Kujala)

Is the manuscript scientifically sound in its present form?

No

Are the interpretations and conclusions justified by the results?

Yes

Is the language acceptable?

Yes

Do you have any ethical concerns with this paper?

No

Have you any concerns about statistical analyses in this paper?

Yes

Recommendation?

Accept with minor revision (please list in comments)

Comments to the Author(s)

The paper considers computation of maximal information coefficient (MIC), defined as

$$\text{MIC}(X,Y) = \max_{\{n_x \times n_y \leq B(n)\}} I(X,Y) / \min(n_x, n_y),$$

where $I(X,Y)$ denotes the maximum mutual information under all possible partitions of the observations into a grid of size $n_x \times n_y$, where $B(n)$ determines the maximum number of grid cells (usually defined as n^α with α 0.6 or 0.55), and where n is the number of observations.

MIC was originally implemented (approximatively) by the ApproxMaxMI algorithm that first divides one axis into equal (by frequency) bins and then finds the optimal grid of a given size for the other axis by a dynamic programming algorithm. The algorithm searches over all possible grid sizes and both orders of using the axes, either x first or y first.

The later ChiMIC algorithm improves on this by replacing the maximum grid size $B(n)$ by a Chi square test which decides whether to keep refining the partition of the second axis. Thus, ChiMIC is no longer trying to maximize the $MIC(X,Y)$ expression under the same constraints, but it is shown to have better properties than ApproxMaxMI.

The idea of the present article is to first take the ChiMIC solution, which gives equipartition for one axis and the optimized partition for the other axis, and then apply the optimization procedure again to the first axis given the optimized solution of the second axis. This is called BackMIC.

Robustness with respect to choice of tuning parameter values, statistical power, and equitability are assessed to show that BackMIC compares favorably to ApproxMaxMI and ChiMIC.

Evaluation of BackMIC as a clustering distance measure and comparison to ApproxMaxMI and ChiMIC as well as to the usual Euclidean distance shows that BackMIC performs better than all the mentioned alternatives in this specific use case for certain datasets. Therefore, BackMIC appears to be useful in practice.

Therefore, I can recommend publication given a revision in which the authors consider the following technical issues:

- this is a very open ended issue, but it would be useful to have some more explicit description of what BackMIC is trying to calculate, for example, if it could be characterized as an estimate of a certain function of the joint distribution of X and Y; presently it is given as just an algorithm whose computed values are shown to behave nicely in certain simulations. Equitability seems to be an implied goal, but BackMIC as well as ChiMIC and ApproxMaxMI are theoretically clearly suboptimal for that goal because a functional dependence with a lot of noise added can have a MIC value of 1.0 - the example being the Heaviside step function + uniform noise.

- the paper describes BackMIC as a backtracking strategy, but if I understood the description correctly, the aspect referred to as "backtracking" is simply a two-step iterative algorithm, not a backtracking algorithm as described in, e.g., <https://en.wikipedia.org/wiki/Backtracking>

- a related issue is what would happen if the iteration was continued for more than two steps - would the MIC values continue to improve and would they converge?

- p. 3, l. 16-19: "To avoid the number of bins on the equipartitioned axis from always reaching $B(n, \alpha)/2$, the ChiMIC algorithm replaces $\log_2 \min(n_x, n_y)$ in the MIC definition with $\log_2 n_{\text{equ}}$, where n_{equ} is the number of bins on the equipartitioned axis and $\log_2 n_{\text{equ}} \geq \log_2 \min(n_x, n_y)$." - I cannot find this change of normalization anywhere in the ChiMIC paper [14]. Also, it seems to me that the equipartitioned axis would in fact not always reach the maximum number of bins as that would in some cases decrease the value of MIC.

- p. 4, Sec. 3.4: the calculation of the statistical power should be explained in more detail: What is the precise test that is performed? What is the null hypothesis? That X and Y are independent? What is the test statistic and what is its distribution under the null assumed to be?

- p. 4, Sec. 3.5: "If a statistic gives similar scores to different functions at the same noise level, then the statistic has the property of equitability [18]" - how is the noise level (indicated as $\text{Noise}(1-R^2)$ in Figure 5) determined?

- the claim "Perfect equitability does not exist [20]" seems to be contested in the paper

"Murrell, B., Murrell, D., & Murrell, H. (2014). R2-equitability is satisfiable. Proceedings of the National Academy of Sciences of the United States of America, 111(21), E2160. <https://doi.org/10.1073/pnas.1403623111>"

- p. 5, l. 32. "The chi-square statistic is defined in Eq. (2)" - equation numbers are missing.

- Figure 8 should have a clear caption detailing the steps (a) - (e).

- Algorithm 1 is difficult to follow. It would be useful to properly indent the code. Instead of saying "equipartition y-axis to n_y rows" this partition should be given a name, e.g., `EquipartitionOfy`. Lines 4 and 11 of the algorithm should both be formulated analogously; there is no backtracking, maximization of $I(X,Y)$ should be specified as given a certain partition on X and a certain partition on Y indicated on the same line.

- it would be really useful if the full source code of the BackMIC algorithm was included as supporting information; even better if implementations of ApproxMaxMI and ChiMIC were also included as well as the code used in all the comparisons made in the paper.

Decision letter (RSOS-201424.R0)

Dear Professor Yuan

On behalf of the Editors, we are pleased to inform you that your Manuscript RSOS-201424 "An improved algorithm for the maximal information coefficient and its application" has been accepted for publication in Royal Society Open Science subject to minor revision in accordance with the referees' reports. Please find the referees' comments along with any feedback from the Editors below my signature.

Please submit your revised manuscript and required files (see below) no later than 7 days from today's (ie 06-Nov-2020) date. Note: the ScholarOne system will 'lock' if submission of the revision is attempted 7 or more days after the deadline. If you do not think you will be able to meet this deadline please contact the editorial office immediately.

Please note article processing charges apply to papers accepted for publication in Royal Society Open Science (<https://royalsocietypublishing.org/rsos/charges>). Charges will also apply to papers transferred to the journal from other Royal Society Publishing journals, as well as papers

submitted as part of our collaboration with the Royal Society of Chemistry (<https://royalsocietypublishing.org/rsos/chemistry>). Fee waivers are available but must be requested when you submit your revision (<https://royalsocietypublishing.org/rsos/waivers>).

on behalf of Professor Ion Petre (Associate Editor) and Mark Chaplain (Subject Editor)
openscience@royalsociety.org

Reviewer comments to Author:
Reviewer: 1

Comments to the Author(s)

The author developed BackMIC algorithm for MIC estimation. It removed the equipartition restriction in grid partition, and terminated the grid search process by the chi-square test. The results showed that BackMIC algorithm gives more accurate MIC values, and improves the robustness, statistical power and equitability of MIC. The application of BackMIC to samples clustering in cancer classification is also valuable.

There are still some problems to be noticed:

1. The BackMIC algorithm removes the equipartition restriction of one axis. What is the axis?
2. the calculation of MIC needs more detailed explanation.
3. At present, many researches are focused on very large data sets, but when comparing the time complexity, the maximum data size is 4000, and the advantages are not obvious. It is recommended to compare on larger data sets.
4. There are a few errors. 1) Where is Eq. (2) in line 32? 2) Figure 2h is a mistake, the author mentioned in line 56 that the grid of BackMIC algorithms for the checkboard data is 5×6 , however, the grid in Figure 2h is 5×5 .
5. Pay attention to English grammar and sentence structure to ensure that the paper can be better understanding.

Reviewer: 2

Comments to the Author(s)

The paper considers computation of maximal information coefficient (MIC), defined as

$$\text{MIC}(X,Y) = \max_{\{n_x \times n_y \leq B(n)\}} I(X,Y) / \min(n_x, n_y),$$

where $I(X,Y)$ denotes the maximum mutual information under all possible partitions of the observations into a grid of size $n_x \times n_y$, where $B(n)$ determines the maximum number of grid cells (usually defined as n^α with α 0.6 or 0.55), and where n is the number of observations.

MIC was originally implemented (approximatively) by the ApproxMaxMI algorithm that first divides one axis into equal (by frequency) bins and then finds the optimal grid of a given size for

the other axis by a dynamic programming algorithm. The algorithm searches over all possible grid sizes and both orders of using the axes, either x first or y first.

The later ChiMIC algorithm improves on this by replacing the maximum grid size $B(n)$ by a Chi square test which decides whether to keep refining the partition of the second axis. Thus, ChiMIC is no longer trying to maximize the $MIC(X,Y)$ expression under the same constraints, but it is shown to have better properties than ApproxMaxMI.

The idea of the present article is to first take the ChiMIC solution, which gives equipartition for one axis and the optimized partition for the other axis, and then apply the optimization procedure again to the first axis given the optimized solution of the second axis. This is called BackMIC.

Robustness with respect to choice of tuning parameter values, statistical power, and equitability are assessed to show that BackMIC compares favorably to ApproxMaxMI and ChiMIC.

Evaluation of BackMIC as a clustering distance measure and comparison to ApproxMaxMI and ChiMIC as well as to the usual Euclidean distance shows that BackMIC performs better than all the mentioned alternatives in this specific use case for certain datasets. Therefore, BackMIC appears to be useful in practice.

Therefore, I can recommend publication given a revision in which the authors consider the following technical issues:

- this is a very open ended issue, but it would be useful to have some more explicit description of what BackMIC is trying to calculate, for example, if it could be characterized as an estimate of a certain function of the joint distribution of X and Y; presently it is given as just an algorithm whose computed values are shown to behave nicely in certain simulations. Equitability seems to be an implied goal, but BackMIC as well as ChiMIC and ApproxMaxMI are theoretically clearly suboptimal for that goal because a functional dependence with a lot of noise added can have a MIC value of 1.0 - the example being the Heaviside step function + uniform noise.

- the paper describes BackMIC as a backtracking strategy, but if I understood the description correctly, the aspect referred to as "backtracking" is simply a two-step iterative algorithm, not a backtracking algorithm as described in, e.g., <https://en.wikipedia.org/wiki/Backtracking>

- a related issue is what would happen if the iteration was continued for more than two steps - would the MIC values continue to improve and would they converge?

- p. 3, l. 16-19: "To avoid the number of bins on the equipartitioned axis from always reaching $B(n, \alpha)/2$, the ChiMIC algorithm replaces $\log_2 \min(n_x, n_y)$ in the MIC definition with $\log_2 n_{\text{equ}}$, where n_{equ} is the number of bins on the equipartitioned axis and $\log_2 n_{\text{equ}} \geq \log_2 \min(n_x, n_y)$." - I cannot find this change of normalization anywhere in the ChiMIC paper [14]. Also, it seems to me that the equipartitioned axis would in fact not always reach the maximum number of bins as that would in some cases decrease the value of MIC.

- p. 4, Sec. 3.4: the calculation of the statistical power should be explained in more detail: What is the precise test that is performed? What is the null hypothesis? That X and Y are independent? What is the test statistic and what is its distribution under the null assumed to be?

- p. 4, Sec. 3.5: "If a statistic gives similar scores to different functions at the same noise level, then the statistic has the property of equitability [18]" - how is the noise level (indicated as $\text{Noise}(1-R^2)$ in Figure 5) determined?

- the claim "Perfect equitability does not exist [20]" seems to be contested in the paper

"Murrell, B., Murrell, D., & Murrell, H. (2014). R2-equitability is satisfiable. Proceedings of the National Academy of Sciences of the United States of America, 111(21), E2160.
<https://doi.org/10.1073/pnas.1403623111>"

- p. 5, l. 32. "The chi-square statistic is defined in Eq. (2)" - equation numbers are missing.

- Figure 8 should have a clear caption detailing the steps (a) - (e).

- Algorithm 1 is difficult to follow. It would be useful to properly indent the code. Instead of saying "equipartition y-axis to n_y rows" this partition should be given a name, e.g., EquipartitionOfy. Lines 4 and 11 of the algorithm should both be formulated analogously; there is no backtracking, maximization of $I(X,Y)$ should be specified as given a certain partition on X and a certain partition on Y indicated on the same line.

- it would be really useful if the full source code of the BackMIC algorithm was included as supporting information; even better if implementations of ApproxMaxMI and ChiMIC were also included as well as the code used in all the comparisons made in the paper.

===PREPARING YOUR MANUSCRIPT===

===PREPARING YOUR REVISION IN SCHOLARONE===

Author's Response to Decision Letter for (RSOS-201424.R0)

See Appendix A.

RSOS-201424.R1 (Revision)

Review form: Reviewer 1

Is the manuscript scientifically sound in its present form?

Yes

Are the interpretations and conclusions justified by the results?

Yes

Is the language acceptable?

Yes

Do you have any ethical concerns with this paper?

No

Have you any concerns about statistical analyses in this paper?

No

Recommendation?

Accept as is

Comments to the Author(s)

I think the manuscript has majorly improved and can now be published by Royal Society Open Science.

Review form: Reviewer 2 (Janne Kujala)

Is the manuscript scientifically sound in its present form?

Yes

Are the interpretations and conclusions justified by the results?

Yes

Is the language acceptable?

Yes

Do you have any ethical concerns with this paper?

No

Have you any concerns about statistical analyses in this paper?

No

Recommendation?

Accept as is

Comments to the Author(s)

The paper has been revised as requested and can now be published.

Decision letter (RSOS-201424.R1)

Dear Professor Yuan,

It is a pleasure to accept your manuscript entitled "An improved algorithm for the maximal information coefficient and its application" in its current form for publication in Royal Society Open Science.

Kind regards,

on behalf of Professor Ion Petre (Associate Editor) and Mark Chaplain (Subject Editor)
openscience@royalsociety.org

Reviewer comments to Author:

Reviewer: 1

Comments to the Author(s)

I think the manuscript has majorly improved and can now be published by Royal Society Open Science.

Reviewer: 2

Comments to the Author(s)

The paper has been revised as requested and can now be published.

Appendix A

Dear Professor Anita Kristiansen:

Thank you very much for giving us the opportunity to revise our manuscript entitled “**An improved algorithm for the maximal information coefficient and its application**” (ID: RSOS-201424.R1). All the suggestions from the editors and reviews are very important, and they are of great guiding significance to my future scientific research.

We have studied review’s comments carefully and have made correction on the original manuscript which we hope meet with approval. All revised portions are marked in red in the revised manuscript which we would like to submit for your kind consideration.

The responds to the reviews’ comments are as follows (the replies are highlighted in blue and some formulas are shown in black).

Replies to Review1:

1. The BackMIC algorithm removes the equipartition restriction of one axis. What is the axis?

Response: MIC is defined as the largest normalized mutual information. To calculate the mutual information of two continuous variables, we need to discretize them by an grid on their scatterplot. When referring to “equipartition of the y-axis”, it means the partition along the y-axis such that each row contains the same number of data points.

For example, X and Y are shown in the following table. When equipartition of the y-axis into 2 bins(rows), the corresponding figure is shown below:

X	0.31	0.6	0.08	0.16	0.26	0.77	0.32	0.75	0.57	0.85	0.77	0.51	0.52	0.74	0.45	0.7	0.16	0.89	0.96	0.61
Y	0.91	0.43	0.64	0.78	0.69	0.6	0.57	0.32	0.27	0.24	0.71	0.76	1	0.02	0.48	0.97	0.25	0.95	0.26	0.45

So, “partition of the axis” in our paper means partition the data points along this axis.

2. The calculation of MIC needs more detailed explanation.

Response: Given a paired variable (X, Y) , $X \in R^n$, $Y \in R^n$. (X, Y) corresponds to n data points on the coordinate axis, and we partitioned these n data points into n_x bins(columns) and n_y bins(rows) along the x -axis and y -axis, respectively. The calculation of $MIC(X, Y)$ is shown as follows:

$$MIC(X, Y) = \max_{n_x \times n_y \leq B(n)} \left\{ \frac{\max_G I_G(X, Y)}{\log_2 \min(n_x, n_y)} \right\}$$

where G represents a $n_x \times n_y$ grid on (X, Y) , $I_G(X, Y)$ denotes the mutual information under the grid G , and its calculation formula is as follow:

$$I_G(X, Y) = H(X) + H(Y) - H(X, Y) = \sum_{i=1}^{n_x} \frac{n_{*,i}}{n} \log \frac{n}{n_{*,i}} + \sum_{j=1}^{n_y} \frac{n_{j,*}}{n} \log \frac{n}{n_{j,*}} - \sum_{i=1}^{n_x} \sum_{j=1}^{n_y} \frac{n_{j,i}}{n} \log \frac{n}{n_{j,i}}$$

where $n_{*,i}$ is the number of data points in the i -th column of the grid G , $n_{j,*}$ is the number of data points in the j -th row of G , $n_{j,i}$ is the number of data points in the i -th column and j -th row of G . $B(n)$ is the maximum number of bins, and $\log_2 \min(n_x, n_y)$ is a normalization term to ensure MIC in the range 0 to 1.

For example, X and Y are the paired variables shown in the table above. Suppose these 20 data points are partitioned into 3 bins ($n_x=3$) and 2 bins ($n_y=2$) along the x -axis and y -axis, respectively, and one of the possible grids G is shown in the figure below.

The mutual information $I_G(X, Y)$ and the normalized mutual information $I_G(X, Y) / \log_2 \min(n_x, n_y)$ achievable by this 3×2 grid G is computed as follows:

Here, $n_{*,1}=3, n_{*,2}=7, n_{*,3}=10; n_{1,*}=8, n_{2,*}=12; n_{1,1}=1, n_{1,2}=1, n_{1,3}=6; n_{2,1}=2, n_{2,2}=6, n_{2,3}=4$.

$$\begin{aligned} I_G(X, Y) &= \sum_{i=1}^{n_x} \frac{n_{*,i}}{n} \log \frac{n}{n_{*,i}} + \sum_{j=1}^{n_y} \frac{n_{j,*}}{n} \log \frac{n}{n_{j,*}} - \sum_{i=1}^{n_x} \sum_{j=1}^{n_y} \frac{n_{j,i}}{n} \log \frac{n}{n_{j,i}} \\ &= \left(\frac{3}{20} \log \frac{20}{3} + \frac{7}{20} \log \frac{20}{7} + \frac{10}{20} \log \frac{20}{10} \right) + \left(\frac{8}{20} \log \frac{20}{8} + \frac{12}{20} \log \frac{20}{12} \right) \\ &\quad - \left(\frac{1}{20} \log \frac{20}{1} + \frac{1}{20} \log \frac{20}{1} + \frac{6}{20} \log \frac{20}{6} + \frac{2}{20} \log \frac{20}{2} + \frac{6}{20} \log \frac{20}{6} + \frac{4}{20} \log \frac{20}{4} \right) \\ &= 0.1406 \end{aligned}$$

$$\frac{I_G(X, Y)}{\log_2 \min(n_x, n_y)} = \frac{0.1406}{\log_2 \min(3, 2)} = \frac{0.1406}{\log_2 2} = 0.1406$$

After traversing all possible grids satisfying $n_x \times n_y \leq B(n)$, the maximal normalized mutual information is the MIC value.

3. At present, many researches are focused on very large data sets, but when comparing the time complexity, the maximum data size is 4000, and the advantages are not obvious. It is recommended to compare on larger data sets.

Response: We compared the running time of BackMIC algorithm and ApproxMaxMI algorithm on datasets with data size $n = 8000$ and 12000 , respectively.

For independent paired variables with data size $n = 8000$ and $n = 12000$, the average consumption time at 100 repetitions by BackMIC algorithm were 16.9615 and 109.7140, while that of ApproxMaxMI algorithm is 22.9444 and 181.9063, respectively. For the parabolic functional variable pairs with noise level of 0.5 with data size $n = 8000$ and $n = 12000$, the average consumption time at 100 repetitions by BackMIC algorithm were 22.4220 and 52.7978, while that of ApproxMaxMI algorithm is 27.2711 and 68.9850, respectively. Therefore, the BackMIC algorithm and the ApproxMaxMI algorithm are close in time-consuming, which is acceptable.

4. There are a few errors. 1) Where is Eq. (2) in line 32? 2) Figure 2h is a mistake, the author mentioned in line 56 that the grid of BackMIC algorithms for the checkboard data is 5×6 , however, the grid in Figure 2h is 5×5 .

Response: We are sorry for these mistakes. 1) We have removed "The chi-square statistic is defined in Eq.(2)" to the end of the second paragraph of **Section 5.1** and changed it to "The chi-square statistic is defined as follow". 2) We also modified the partition of the checkerboard data in Figure 2h under the BackMIC algorithm to the 5×6 grid.

5. Pay attention to English grammar and sentence structure to ensure that the paper can be better understanding.

Response: Thank you for your advice. We carefully revised our paper, and invited emerald, an English-language editing service, to polish our wording to improve its readability. The certification of editing is as follow:

Replies to Review2:

1. This is a very open ended issue, but it would be useful to have some more explicit description of what BackMIC is trying to calculate, for example, if it could be characterized as an estimate of a certain function of the joint distribution of X and Y . Presently it is given as just an algorithm whose computed values are shown to behave nicely in certain simulations. Equitability seems to be an implied goal, but BackMIC as well as ChiMIC and ApproxMaxMI are theoretically clearly suboptimal for that goal because a functional dependence with a lot of noise added can have a MIC value of 1.0 - the example being the Heaviside step function + uniform noise.

Response: 1) BackMIC is an approximate estimation algorithm for MIC. MIC is the correlation measure of paired variables which is defined as the largest normalized mutual information achieved by any $n_x \times n_y$ grid applied to the variable pair.

The mutual information of two discrete random variables of X and Y is defined as:

$$I(X, Y) = \sum_{y \in Y} \sum_{x \in X} p(x, y) \log \frac{p(x, y)}{p(x)p(y)}$$

Where $p(x, y)$ is the joint distribution of the pair (X, Y) , $p(x)$ and $p(y)$ are the marginal distributions of X and Y . The definition of mutual information shows that, the mutual information determines how different the joint distribution $p(x, y)$ is to the product of the marginal distributions $p(x)$ and $p(y)$. Therefore, MIC can be regarded as the normalized difference between the joint distribution $p(x, y)$ and the product of the marginal distributions $p(x)$ and $p(y)$.

2) Equitability is indeed a controversial issue. In our paper, we only used several no-where constant functions similar to those used in Reshef et al. ^[1], to compare the equitability of the MIC obtained by the three algorithms. Heaviside step function as a piecewise constant function seems not satisfy the equitability notion. A typical Heaviside step + uniform noise is shown below ^[4], the MIC equals 1 which may due to that we generally use grids parallel to the x -axis and y -axis to discretize the data.

2. The paper describes BackMIC as a backtracking strategy, but if I understood the description correctly, the aspect referred to as "backtracking" is simply a two-step iterative algorithm, not a backtracking algorithm as described in, e.g., <https://en.wikipedia.org/wiki/Backtracking>

Response: We are very glad to accept your suggestion. As you mentioned, the "backtracking" here is essentially iterative. In fact, backtracking means that in the process of searching for the optimal solution, when it is found that the candidate cannot meet the valid solution, it will abandon the candidate and reselect a new one.

In our paper, we mean that, when searching for the optimal partition on the scatterplot of two variables, we first equipartitioned one axis, and searched for the optimal partition of the other axis, then searched again for the optimal partition of the first axis based on the optimal partition of the second axis. Therefore, we changed the term "backtracking" in our paper to "searching back" to indicate the two-step iterative process in the BackMIC algorithm.

3. A related issue is what would happen if the iteration was continued for more than two steps - would the MIC values continue to improve and would they converge?

Response: MIC is the largest normalized mutual information. However, under the limitation of $B(n)$ ($B(n) = n^{0.55}$ or $n^{0.6}$), the computational cost of exhausting all grids to obtain MIC is too high. Therefore, the BackMIC algorithm searches for an approximate optimal grid and obtains the approximate MIC value through the dynamic programming process. In theory, the more iterations, the more grids are searched, and the more likely it is to approach the optimal grid. However, on the other hand, the more iterations, the more time the algorithm consumes, which means that "efficiency and precision cannot be achieved at the same time". We hope to obtain accurate MIC estimation within acceptable time complexity.

Taking the noise linear, parabolic and sinusoidal correlation in Figure 3 of our manuscript as an example, we compare the MIC values obtained by BackMIC algorithm in 2-10 iterations (the result is the average MIC value at 100 repetitions). As can be seen from the figure below, as the number of iterations increases, although the MIC value has no obvious convergence trend, the change of MIC value is not obvious. This shows to a certain extent that it is feasible for the BackMIC algorithm to iterate only twice. The results show that the MIC value decreases occasionally, which may be caused by the BackMIC algorithm falling into local optimal solution when using the dynamic programming algorithm to find the optimal grid.

4. 1) P3, 1. 16-19: "To avoid the number of bins on the equipartitioned axis from always reaching $B(n, \alpha)/2$, the ChiMIC algorithm replaces $\log_2 \min(n_x, n_y)$ in the MIC definition with $\log_2 n_{\text{equ}}$, where n_{equ} is the number of bins on the equipartitioned axis and $\log_2 n_{\text{equ}} \geq \log_2 \min(n_x, n_y)$." - I cannot find this change of normalization anywhere in the ChiMIC paper [2]. 2) Also, it seems to me that the equipartitioned axis would in fact not always reach the maximum number of bins as that would in some cases decrease the value of MIC.

Response: 1) Although the ChiMIC algorithm changes the normalization item in MIC definition, it is not explicitly mentioned in the article [2].

MIC is the largest normalized mutual information. In general, computing the mutual information of two continuous random variables X and Y always requires discretization of them, the higher discretization degree of X and Y , the larger the mutual information $I(X, Y)$. Therefore, from

the formula
$$MIC(X, Y) = \max_{n_x \times n_y \leq B(n, \alpha)} \left\{ \frac{I_G(X, Y)}{\log_2 \min(n_x, n_y)} \right\}$$
, the MIC value always obtained when $n_x \times n_y$

reaches $B(n)$. This leads to the generality of MIC closely related to $B(n)$, if a low $B(n)$ is set, the MIC can only capture simple correlation patterns; by contrast, a high $B(n)$ will cause a non-zero score even for independent variables [1]. For example, when data size is 100, MIC value calculated by ApproxMaxMI is 0.24 while the optimal grid is 2×8 or 8×2 . The key to solve this problem is to reasonably control the number of bins along the x -axis and y -axis, so as to avoid obtaining the MIC value when the bin number reaches $B(n)$.

ChiMIC used the chi-square test instead of $B(n)$ to control the number of bins along the optimization axis. However, ChiMIC does not limit the number of bins along the equipartitioned axis, which still makes MIC tend to be obtained when $n_x \times n_y$ reaches $B(n)$. For example, when we used the ChiMIC algorithm to calculate the MIC of an independent variable pair with data size $n = 100$ (assuming that the y -axis is equipartitioned), if $\log_2 \min(n_x, n_y)$ is still used as the normalization term (left in the figure below), the optimal grid tends to be 8×2 (y -axis is equipartitioned into 8 bins, x -axis is partitioned into 2 bins). Because under the limitation of $B(n) = \text{ceil}(100^{0.6}) = 16$, the discretization degree of X and Y is the highest ($2 \times 8 = 16$), the mutual information $I_G(X, Y)$ is more

likely to be the highest. Meanwhile, the normalization term $\log_2 \min(n_x, n_y)$ is the smallest ($\log_2 \min(n_x, n_y) = \log_2 \min(2, 8) = 1$), then, the MIC value is obtained

$$(MIC(X, Y) = \max_{n_x \times n_y \leq B(n)} \left\{ \frac{I(X, Y)}{\log_2 \min(n_x, n_y)} \right\} \leq \frac{\max_{n_x \times n_y \leq B(n)} I(X, Y)}{\min(\log_2 \min(n_x, n_y))} = \max_{n_x \times n_y \leq B(n)} I(X, Y)).$$

Therefore, the

MIC value obtained by ChiMIC still obtained when $n_x \times n_y$ reaches $B(n)$. For this reason, ChiMIC replaces $\log_2 \min(n_x, n_y)$ with a harsher normalization term $\log_2 n_{equ}$ (right in the figure below) to control the number of bins along the equipartitioned axis.

	2	3	4	5	6	7	8
2	log2	log2	log2	log2	log2	log2	log2
3	log2	log3	log3	log3			
4	log2	log3	log4				
5	log2	log3					
6	log2						
7	log2						
8	log2						

	2	3	4	5	6	7	8
2	log2	log2	log2	log2	...		
3	log3	log3	log3	...			
4	log4	log4	log4				
5	log5	log5					
6	log6						
7	log7						
8	log8						

The first column of the above tables represents the number of bins along the equipartitioned y-axis, and the first row represents the number of bins along the x-axis for optimization.

2) Indeed, more bins on the equipartitioned axis may not always achieve larger MIC. However, in general, the mutual information with more bins on the equipartitioned axis (such as the y-axis) is no smaller than that with fewer bins when the partition of the x-axis is the same and $n_x \leq n_y$ ($\log_2 \min(n_x, n_y) = \log_2 n_x$).

5. P4, Sec. 3.4: the calculation of the statistical power should be explained in more detail: What is the precise test that is performed? What is the null hypothesis? That X and Y are independent? What is the test statistic and what is its distribution under the null assumed to be?

Response: The power of a statistical test of a null hypothesis (H_0) is the probability that the H_0 will be rejected when it is false, that is $1 - \beta$, β is the probability of making type II error^[3].

In our paper, the null hypothesis (H_0): a paired variable is independent. So, the statistical power is the probability to correctly detect the variable pairs with correlation. Detecting correlated variable pairs with noise by different statistics, the higher the detection rate is, the higher the statistical power of the statistic is.

The calculation process of statistical power is as follows:

For each dataset, statistical power is computed on the dependent variable pairs as well as on independent variable pairs, the statistical power of each statistic is defined as the fraction of dependent variable pairs yielding a statistic value greater than 95% (significance level is 0.05) of the values yielded by the independent variable pairs.

The above calculation process is added to **Section 3.4** of our manuscript.

6. P4, Sec. 3.5: "If a statistic gives similar scores to different functions at the same noise level, then the statistic has the property of equitability [18]" - how is the noise level (indicated as $\text{Noise}(1-R^2)$ in Figure 5) determined?

Response: For a random variable X and a function f of X , defining $Y = f(X) + \eta$. If η is a random variable drawn from $(-b, b)$, we called b the "noise level".

The expression of "noise level" in equitability definition in Section 3.5 and the caption of Figure 5 are ambiguous. So, first, we have changed the definition of equitability as "If a statistic assigns similar scores to equally noisy correlations of different types, then the statistic has the property of equitability", where "noise" quantified by $1 - R^2(f(X), Y)$, R^2 is the squared Pearson correlation coefficient of $f(X)$ and Y . Second, we changed the caption of Figure 5 as "**Figure 5. Statistical power of AppMIC, ChiMIC, and BackMIC at different noise amplitudes.** The statistical power was estimated via 500 simulations with data size $n = 500$." The noise amplitude a for the statistical power calculation in Figure 5, is 25 noise amplitudes with logarithmic distribution from 1 to 10(explained in **Section 4.1 Simulated data**).

7. The claim "Perfect equitability does not exist" seems to be contested in the paper. "Murrell, B., Murrell, D., & Murrell, H. (2014). R^2 -equitability is satisfiable. Proceedings of the National Academy of Sciences of the United States of America, 111(21), E2160. <https://doi.org/10.1073/pnas.1403623111>"

Response: The controversy over equitability always exists [4-7]. Equitability means that "the statistic should give similar scores to equally noisy relationships of different types", this heuristic notion of equitability was put forward by Reshef *et al* [1], and they also pointed out that the equitability of MIC is better than other statistics such as mutual information. And the notion of equitability is not a strict mathematical definition. For example, it does not define what extent can be considered "similar", nor how to add noise, how to determine whether the noise is equal, and so on. So, Kinney and Atwal [4] gave a formalization criterion of equitability based on the heuristic notion of equitability, called R^2 -equitability, and stated that no nontrivial dependence measure can satisfy R^2 -equitability, including MIC. Simon and Tibshirani [8] pointed out through numerical simulation that the statistical power of MIC is lower than that of some other statistics, and further pointed out that it is meaningless to define "equitability" under such low statistical power.

In response to the paper of Kinney and Atwal, Murrell *et al.* [5] argued that "no nontrivial dependence measure can satisfy R^2 -equitability" is due to the poorly constructed definition of equitability. For $Y = f(X) + \eta$, when the noise term η in R^2 -equitability definition changes from "The noise term η may depend on $f(X)$ as long as η has no additional dependence on X " to "allowing η to depend arbitrarily on $f(X)$ ", some measures could satisfy the R^2 -equitability. Reshef *et al.* [6] point

out that R^2 -equitability definition is different from the heuristic notion of equitability, the former would require that MIC exactly equal R^2 in the infinite data limit. So, the proof that no dependence measure can satisfy R^2 -equitability does not uncover any error in their work.

Meanwhile, although it does not make much sense to discuss equitability under a lower statistical power, we have greatly improved the statistical power of MIC by optimizing MIC estimation algorithm. Therefore, the discussion of equitability is valuable.

8. P5, l. 32. "The chi-square statistic is defined in Eq. (2)" - equation numbers are missing.

Response: We are sorry for the mistake, and moved "The chi-square statistic is defined in Eq.(2)" to the end of the second paragraph of **Section 5.1** and changed it to "The chi-square statistic is defined as follow:".

9. Figure 8 should have a clear caption detailing the steps (a) - (e).

Response: In order to better understand the process of the BackMIC algorithm, we moved the description of Figure 8 from Section 5.2 to the caption of Figure 8. The caption of Figure 8 is modified as:

Figure 8. Simulation process of the BackMIC algorithm. (a) The scatterplot of a paired variable with data size $n = 40$; (b) Equipartition the y -axis into 2 bins; (c) Search for the optimal partition on the x -axis to obtain the largest normalized mutual information $M_y(2)=0.3113$; (d) Fix the partition of the x -axis obtained in (c); (e) Search back for the optimal partition on the y -axis to obtain the largest normalized mutual information $M_{Back_y}(2)= 0.5414$.

10. Algorithm 1 is difficult to follow. It would be useful to properly indent the code. Instead of saying "equipartition y -axis to n_y rows" this partition should be given a name, e.g., EquipartitionOf y . Lines 4 and 11 of the algorithm should both be formulated analogously; there is no backtracking, maximization of $I(X,Y)$ should be specified as given a certain partition on X and a certain partition on Y indicated on the same line.

Response: As you mentioned, we first indented the pseudo code of the BackMIC algorithm, then we changed "backtracking" to "searching back", expressed "equipartition y -axis" as "EquipartitionOf y ", "search for the (n_x-1) -th segment point SP_{n_x} on the x -axis" as "search for SegPointOf x_{n_x} ", and "search for the $(s-1)$ -th segment point SP_s on the y -axis" as "search for SegPointOf y_s ".

To clarify that $I(X, Y)$ should be specified as a certain given partition on X and Y , we first changed the definition of MIC in "Section 2 Introduction" from

$$MIC(X, Y) = \max_{n_x \times n_y \leq B(n, \alpha)} \left\{ \frac{I(X, Y)}{\log_2 \min(n_x, n_y)} \right\} \text{ to } MIC(X, Y) = \max_{n_x \times n_y \leq B(n, \alpha)} \left\{ \frac{\max_G(I_G(X, Y))}{\log_2 \min(n_x, n_y)} \right\}, \text{ where } G$$

represents a $n_x \times n_y$ grid on (X, Y) , $I_G(X, Y)$ denotes the mutual information under the grid G . Then the mutual information $I(X, Y)$ in Algorithm 1 is changed into $I_G(X, Y)$.

The algorithm 1 have been modified as follows:

Algorithm 1 BackMIC algorithm

Input: $(X, Y) = \{(x_1, y_1), (x_2, y_2), \dots, (x_n, y_n)\}$ is a pair of ordered variables, $B=n^{0.6}$, threshold=0.01

```

1   for  $n_y=2$  to ceil( $B/2$ )
2       EquipartitionOfy into  $n_y$  rows; PartionOfx  $\leftarrow \varphi$ ; BackPartionOfy  $\leftarrow \varphi$ 
3       for  $n_x=2$  to  $n$ 
4           search for SegPointOf $x_{n_x}$  maximizing  $I_G(X, Y)/\log_2 \min(n_x, n_y)$ ;
5           Obtain  $\chi^2_{\text{SegPointOf}x_{n_x}}$  and  $p\text{-value}_{\text{SegPointOf}x_{n_x}}$ ;
6           if  $p\text{-value}_{\text{SegPointOf}x_{n_x}} > \text{threshold}$  then exit for end if;
7           PartionOfx  $\leftarrow$  PartionOfx  $\cup$  { SegPointOf $x_{n_x}$  };
8       end for
9        $M_y(n_y) \leftarrow I_G(X, Y)/\log_2 \min(\text{length}(\text{PartionOfx})+1, n_y)$ ;
10      for  $s=2$  to  $n$ 
11          Fix PartionOfx, search back for SegPointOf $y_s$  maximizing  $I_G(X, Y)/\log_2 \min(s, \text{length}(\text{PartionOfx})+1)$ ;
12          Obtain  $\chi^2_{\text{SegPointOf}y_s}$  and  $p\text{-value}_{\text{SegPointOf}y_s}$ ;
13          if  $p\text{-value}_{\text{SegPointOf}y_s} > \text{threshold}$  then exit for end if;
14          BackPartionOfy  $\leftarrow$  BackPartionOfy  $\cup$  {SegPointOf $y_s$ };
15      end for
16       $M_{\text{Back}_y}(n_y) \leftarrow I_G(X, Y)/\log_2 \min(\text{length}(\text{BackPartionOfy})+1, \text{length}(\text{PartionOfx})+1)$ ;
17  end for
18  Switch the axes, repeat the above steps, and obtain  $M_x$  and  $M_{\text{Back}_x}$ .
Output MIC = max { $M_x, M_{\text{Back}_x}, M_y, M_{\text{Back}_y}$ }

```

11. It would be really useful if the full source code of the BackMIC algorithm was included as supporting information; even better if implementations of ApproxMaxMI and ChiMIC were also included as well as the code used in all the comparisons made in the paper.

Response: We added the download address of the BackMIC algorithm in the part of “**Data Accessibility**”, which is described as “An implementation of the BackMIC algorithm are stored in GitHub: <https://github.com/Caodan82/BackMIC>, and have been archived within the Zenodo repository: https://zenodo.org/record/4280173#.X7X3z_nDpw8(DOI:10.5281/zenodo.4280173)”. We also added the download addresses for ApproxMaxMI and ChiMIC where these two algorithms first appeared (in the third paragraph of **Section 2 Introduction**).

In addition, we are sorry that in Table 6 of section 4.2, we mistakenly entered the number of samples and genes in the GSE29272 dataset, which we have corrected in our manuscript.

Kind regards.

Corresponding author: Zheming Yuan

E-mail address: zhmyuan@sina.com

- [1] Reshef DN, Reshef YA, Finucane HK, Grossman SR, McVean G, Turnbaugh PJ, et al. Detecting Novel Associations in Large Data Sets. *Science*. 334(6062):1518-1524. (DOI: 10.1126/science.1205438)
- [2] Chen Y, Zeng Y, Luo F, Yuan Z. A new algorithm to optimize maximal information coefficient. *PloS One*. 2016;11(6). (DOI:10.1371/journal.pone.0157567)
- [3] Cohen J. Statistical power analysis. *Current Directions in Psychological Science*. 1992;1(3):98-101. (DOI: 10.1111/1467-8721.ep10768783)
- [4] Kinney JB, Atwal GS. Equitability, mutual information, and the maximal information coefficient. *Proceedings of the National Academy of Sciences of the United States of America*. 2014;111(9):3354-3359. (DOI: 10.1073/pnas.1309933111)
- [5] Murrell B, Murrell D, Murrell H. R^2 -equitability is satisfiable. *Proceedings of the National Academy of Sciences*. 2014;111(21):E2160.(DOI: 10.1073/pnas.1403623111)
- [6] Reshef DN, Reshef YA, Mitzenmacher M, Sabeti PC. Cleaning up the record on the maximal information coefficient and equitability. *Proceedings of the National Academy of Sciences*. 2014;111(33):E3362-E3363. (DOI:10.1073/pnas.1408920111)
- [7] Reshef YA, Reshef DN, Sabeti PC, Mitzenmacher M. Equitability, Interval Estimation, and Statistical Power. *Statistical Science*. 2020;35(2):202-217. (DOI:10.1214/19-sts719)
- [8] Simon N, Tibshirani R. Comment on" detecting novel associations in large data sets" by reshef et al, *science* dec 16, 2011. arXiv preprint arXiv:14017645. 2014.